# Evaluation of Left Ventricular Function in Healthy Retrievers Using Standard and 2D Speckle-Tracking Echocardiography

**DOI:** 10.3390/vetsci9100529

**Published:** 2022-09-27

**Authors:** Lina Hamabe, Kazumi Shimada, Ahmed S. Mandour, Tomohiko Yoshida, Miki Hirose, Hanan Hendawy, Hussein M. El-Husseiny, Ryou Tanaka

**Affiliations:** 1Department of Veterinary Surgery, Faculty of Veterinary Medicine, Tokyo University of Agriculture and Technology, Tokyo 183-8509, Japan; 2Department of Animal Medicine (Internal Medicine), Faculty of Veterinary Medicine, Suez Canal University, Ismailia 41522, Egypt; 3Department of Veterinary Surgery, Faculty of Veterinary Medicine, Suez Canal University, Ismailia 41522, Egypt; 4Department of Surgery, Anesthesiology and Radiology, Faculty of Veterinary Medicine, Benha University, Benha 13736, Egypt

**Keywords:** echocardiography, Retrievers, speckle tracking, strain

## Abstract

**Simple Summary:**

Standard echocardiography is an essential diagnostic tool for the assessment of cardiac performance in healthy and diseased animals. Two-dimensional speckle-tracking echocardiography is an advanced echocardiographic technique that is useful for the assessment of myocardial function. Factors such as dog breed, age, and body weight (BW) are known to influence the echocardiographic parameters. This study aimed to evaluate the effect of breed, age, and BW on the echocardiographic parameters in healthy Retrievers, including Flat-coated, Golden, and Labrador Retrievers. The results showed that there were no significant associations between breed and echocardiographic parameters. On the other hand, they revealed that left ventricular dimensions increased with BW. Furthermore, parameters including peak aortic blood flow velocity, trans-mitral rapid ventricular filling flow, the ratio of trans-mitral rapid ventricular filling flow to atrial contraction, and global radial strain were influenced by age.

**Abstract:**

Standard echocardiography is vital for the assessment of cardiac performance in healthy and diseased animals. Similarly, two-dimensional speckle-tracking echocardiography (2D-STE) is an advanced echocardiographic technique that is becoming increasingly important for the assessment of myocardial function. Breeds, age, and body weight (BW) are known to be important factors affecting the echocardiographic parameters; therefore, the aim of this study was to evaluate the effect of breed, age, and BW on the echocardiographic parameters in three breeds of clinically healthy Retrievers. A total of 46 Retrievers, including 16 Flat-coated Retrievers (FR), 16 Golden Retrievers (GR), and 14 Labrador Retrievers (LR) were included in the study. The comparison of the breeds revealed significant differences in the LV wall thickness of FR and GR, although further analysis using MLR showed that the differences were most likely associated with BW, similarly to the other LV dimensions. Functional parameters, including ejection fraction, fractional shortening, and left-atrial-to-aortic ratio, were independent of breed, age, and BW. On the other hand, peak aortic blood flow velocity, trans-mitral rapid ventricular filling flow, and the ratio of trans-mitral rapid ventricular filling flow to atrial contraction were influenced by age. The 2D-STE-derived radial and circumferential strain parameters were independent of breed, age, and BW, except for global strain in the radial direction.

## 1. Introduction

Standard echocardiography is a fundamental diagnostic imaging tool for the assessment of cardiac function in veterinary medicine. It allows cardiac morphology, movement of myocardium and valves, and blood flows within the heart to be visualized [1]. In addition to the qualitative assessment, the quantitative assessment of echocardiography allows a non-invasive evaluation of the cardiac function to be performed [1,2]. Not only is it vital for the assessment of cardiac performance in healthy animals, but it is also essential for the diagnosis of cardiac diseases, the evaluation of treatment, and the determination of prognosis. Two-dimensional STE is an advanced echocardiographic technique that assesses myocardial function using deformation imaging [3]. This deformation analysis provides information on regional and global myocardial function and is increasingly adopted in veterinary medicine [3].

Echocardiographic parameters such as cardiac chamber size and wall thickness in dogs appear to be linearly or logarithmically correlated with BW, and normal reference values based on BW have been established [1,2]. However, these reference values are broad due to wide variations in canine body size and body conformation, limiting their application in routine clinical examination [2,4,5]. Additionally, the breed is known to be an important factor affecting the echocardiographic parameters, suggesting the importance of establishing breed-specific reference values [5,6]. Retrievers are a type of gun dogs, initially bred for their ability to retrieve hunting prey. Retrievers’ good nature and intelligence have made breeds such as FR, GR, and LR popular breeds as companion dogs. Retrievers are known to mature to be over 20 kg in weight, and they are categorized as the large breed dogs. These large breed dogs are most predisposed to dilated cardiomyopathy (DCM), and it has been reported that over 95% of dogs diagnosed with DCM weigh over 15 kg [7,8]. Additionally, congenital heart diseases such as sub-valvular aortic stenosis, and tricuspid valve dysplasia and Ebstein’s anomaly are overrepresented in GR and LR, respectively [9,10,11,12]. Therefore, it is important to investigate the normal echocardiographic parameters for healthy Retrieves.

The aim of the present study was to obtain the measurements of standard echocardiographic parameters and 2D-STE parameters from clinically healthy Retrievers, including FR, GR, and LR, and to determine the effect of breed, age, and BW on the echocardiographic parameters.

## 2. Materials and Methods

### 2.1. Animals

Forty-six clinically healthy dogs of three Retriever breeds (FR, GR, and LR) were evaluated prospectively at the Animal Medical Centre at Tokyo University of Agriculture and Technology. This study was conducted in accordance with the standards established by Tokyo University of Agriculture and Technology, and informed consent was obtained from all owners. Dogs were selected based on unremarkable history and normal physical examination, with no evidence of congenital or acquired cardiac disease upon standard echocardiographic examination.

### 2.2. Echocardiography

Standard echocardiographic examination, consisting of 2D, M-mode, spectral Doppler, and pulse-wave tissue Doppler imaging (TDI) assessments, and 2D-STE examination were performed with ALOKA prosound α10 (Hitachi Aloka Medical, Ltd., Tokyo, Japan) using a 3.0–8.0 MHz phased array transducer probe (UST52108; Hitachi Aloka Medical, Ltd., Tokyo, Japan). Examinations were performed in a dark, quiet room. Dogs were loosely restrained in both right and left lateral recumbencies without sedation on an echocardiographic table. For the duration of the examinations, a lead II electrocardiogram was simultaneously recorded. In an effort to reduce variability, echocardiographic examination was performed by a single observer.

### 2.3. Standard Echocardiography

Echocardiographic measurements were made in accordance with the methodology published in the veterinary literature [1,6]. M-mode measurements were made using the leading-edge-to-leading-edge method, with left ventricular (LV) dimensional measurements, including interventricular septal (IVS) thickness at end-diastole (IVSd) and at end-systole (IVSs), LV internal diameter (LVID) at end-diastole (LVIDd) and at end-systole (LVIDs), and LV free wall (LVFW) thickness at end-diastole (LVFWd) and at end-systole (LVFWs), being obtained via the right parasternal short-axis view at the level of chordae tendineae. Diastolic and systolic LVID normalized to BW (LVIDdN and LVIDsN) were calculated as reported by Cornell et al. [13]. End-diastolic volume (EDV) and end-systolic volume (ESV) were calculated according to the Teichholz formula using measurements in M-mode. The following parameters were calculated: ejection fraction (EF) (%) = EDV−ESV/EDV × 100 and fractional shortening (FS) (%) = LVIDd−LVIDs/LVIDd × 100. The left-atrial-to-aortic ratio (LA/Ao) was measured in early diastole via the right parasternal short-axis view at the aortic valve level.

Using the spectral Doppler assessment, peak pulmonary blood flow velocity (PA Vmax) and peak aortic blood flow velocity (Ao Vmax) were obtained via the right parasternal short-axis view at the levels of pulmonary artery and left parasternal apical five-chamber view, respectively. The left parasternal apical four-chamber view was used to record the trans-mitral flow, where rapid ventricular filling flow (E) and atrial contraction (A) were measured and the E/A ratio was calculated.

For the pulse-wave TDI assessment, the left parasternal apical four-chamber view was used to measure the mitral annular tissue velocities. Early diastolic tissue velocity (E′) was obtained at IVS and LVFW, and the corresponding ratio between E and E′ (E/E′) at IVS and LVFW were calculated. Additionally, the TDI-derived myocardial performance indices (MPIs) of IVS and LVFW were calculated using the following formula: TDI-derived MPI = IRT + ICT/LVET. The isovolumetric contraction time (ICT) was measured from the end of late diastolic tissue velocity (A′) to the beginning of systolic motion (S′); isovolumetric relaxation time (IRT) was measured from the end of S′ to the beginning of E′; and LVET was measured from the onset of flow to the end of flow at the baseline of S′.

### 2.4. Two-Dimensional Speckle-Tracking Echocardiography

A right parasternal short-axis view was acquired at the level of the papillary muscle at a frame rate of 70–110 frames/second and was then analyzed off-line. The endocardial and epicardial borders of the LV were manually traced at end-systole by placing several regions of interests (ROIs) at the myocardium. Software, then, automatically tracked the ROIs on a frame-by-frame basis, and strain measurements were taken in the radial and circumferential directions. Strain parameters including global peak strain in the radial (Figure 1) and circumferential (Figure 2) directions (GRS and GCS) and the synchrony time index (STI) in the radial direction were obtained.

### 2.5. Statistical Methods

Statistical analyses were performed using commercial statistical software (Prism 8.0v; GraphPad Software Inc., San Diego, CA, USA). A mean of five measurements was obtained from consecutive cardiac cycles in sinus rhythm for each echocardiographic parameter, and normal distribution was graphically inspected and tested using the Shapiro–Wilk test. The differences in the echocardiographic parameters of each breed of Retrievers were evaluated using the Kruskal–Wallis one-way analysis of covariance followed by Dunn’s multiple comparison test for standard echocardiography. Spearman’s correlation and multiple linear regression (MLR) analyses were performed on the echocardiographic parameters as dependent variables, and age and BW as independent variables. Significant differences were defined as *p* < 0.05.

## 3. Results

A total of 46 clinically healthy Retrieves, including 16 FRs, 16 GRs, and 14 LRs, were included in this study. For FRs, 10 were males and 6 were females. The median age was 3.50 years, ranging from 1 to 11 years, and the mean BW was 26.42 ± 2.93 kg, ranging from 20.00 to 32.85 kg. GRs included seven males and nine females. The median age was 7.50 years, ranging from 1 to 11 years, and the mean BW was 28.87 ± 5.20 kg, ranging from 18.30 to 38.60 kg. For LRs, six were males and eight were females. The median age was 7.00 years, ranging from 1 to 12 years, and the mean BW was 27.33 ± 5.83 kg, ranging from 17.30 to 36.80 kg. Age and BW did not show any significant difference among the breeds (*p* = 0.32 and 0.27 for age and BW, respectively).

### 3.1. Standard Echocardiography

The standard echocardiographic data obtained via 2D, M-mode, spectral Doppler, and pulse-wave TDI echocardiography for the three individual Retriever breeds and overall are presented in Table 1 and Table 2. LV wall thicknesses, including diastolic and systolic IVS and LVFW, showed significant difference between FRs and GRs (*p* = 0.0043 and 0.0008 for IVSd and IVSs, respectively, and *p* < 0.0001 for both LVFWd and LVFWs). Significant differences were also seen between FRs and LRs for IVSs (*p* = 0.0180), and between GRs and LRs for LVFWd (*p* = 0.0253). Additionally, LVIDsN differed significantly between FR and GR (*p* = 0.0214).

Age showed a significant and moderate correlation with PA Vmax (r = −0.54, *p* = 0.0002) and E (r = −0.4755, *p* = 0.0008) and correlated significantly but weakly with LVFWd (r = 0.3174, *p* = 0.0316) and E/A (r = −0.39, *p* = 0.070). BW showed significant correlations with M-mode measurements, apart from LVFWd and LVFWs. Moderate correlations were observed with IVSs (r = 0.43, *p* = 0.0030) and LVIDd (r = 0.42, *p* = 0.0039) and weak correlations with IVSd (r = 0.31, *p* = 0.0386) and LVIDs (r = 0.37, *p* = 0.0117).

MLR analyses revealed significant associations between age and parameters, including PA Vmax (*p* = 0.0010, β = −2.16, 95% CI of β = −3.39 to −0.93), E (*p* = 0.0024, β = −1.89, 95% CI of β = −3.07 to −0.71), and E/A (*p* = 0.0047, β = −0.04, 95% CI of β = −0.07 to −0.01), and significant associations between BW and parameters, including IVSs (*p* = 0.0231, β = 0.14, 95% CI of β = 0.02 to 0.25), LVIDd (*p* = 0.0023, β = 0.29, 95% CI of β = 0.11 to 0.47), LVIDs (*p* = 0.0051, β = 0.24, 95% CI of β = 0.08 to 0.40), and A (*p* = 0.0492, β = −0.59, 95% CI of β = −1.18 to −0.002).

### 3.2. Two-Dimensional Speckle-Tracking Echocardiography

Two-dimensional STE examination was performed in 28 dogs out of the 46 Retrievers. The two-dimensional STE parameters for these three Retriever breeds and overall are presented in Table 3 and Table 4. The strain parameters did not show any significant difference among the breeds. Age revealed a significant and moderate correlation with global radial strain (GRS) (r = −0.4736, *p* = 0.0109), and MLR analyses also revealed a significant association between age and GRS (*p* = 0.0119, β = −0.49, 95% CI of β = −0.86 to −0.12).

## 4. Discussion

The present study evaluated the effect of breed, age, and BW on the parameters of standard echocardiography and 2D-STE obtained from clinically healthy Retrievers. FR, GR, and LR are recognized as separate breeds, but are closely related based on genetic and phenotypic breed groupings, belonging to the gun-dog, with similar deep-chested conformation [14].

It is well documented that LV dimensions increase with BW, and similar results were obtained in this study, apart from LVFW [2,5,6]. In terms of breed differences, a study by Morrison et al., which compared the echocardiographic parameters of four morphologically different breeds (Pembroke Welsh Corgi, Miniature Poodle, Afghan Hound, and GR), showed significant breed variations for LV dimensions despite considering the differences in BW [5]. Such finding suggests that LV dimensions may be affected by body conformation independently of BW [5]. This study, which investigated breeds with similar body conformation, revealed significant differences in the LV wall thickness of FR and GR. However, further analysis using MLR suggested that the differences were most likely associated with BW rather than breed. Similar results were reported by della Torre et al., who compared the echocardiographic parameters of sighthounds with varying BW but with identical body conformation (Greyhounds, Whippets, and Italian Greyhounds), revealing that the LV dimensions were correlated with body size [4].

In this study, functional parameters such as EF and FS did not show any significant association with breed, age, or BW. In the veterinary literature, there are mixed reports on the associations between BW and the functional parameters, where some reported the latter to be independent of BW [15,16,17,18,19,20,21], while others revealed a negative correlation [4,19,22,23,24,25]. The results of this study and others showed no significant correlations between the functional parameters and age, which suggests that decreased FS is not a normal aging process [24]. LA/Ao is known to give a body-size-independent measurement of LA, and LA/Ao in this study was not influenced either by breed, age, or BW [21,22,23,26].

While the results of spectral Doppler-derived and pulse-wave-TDI-derived parameters did not show any variations with breed or BW, they revealed significant associations between age, and PA Vmax, E, and E/A. Similarly, a study by Wehrum et al. evaluated the pulmonary artery blood flow within three different age-groups (20–39, 40–59, and 60–80 years of age) in the general human population using 4D flow cardiovascular magnetic resonance, and lower pulmonary flow velocities were observed in the older than in the younger subjects, resulting from an age-related reduction in cardiac function and vascular elasticity [27]. A similar result was also documented in dogs [24]. Age is also known to be a major factor in altering the trans-mitral flow profile in both humans and dogs [1,26,28]. The early filling of the LV is influenced by changes in relaxation, while the late diastolic filling of the LV is influenced by changes in compliance, and this decrease in cardiac function is observed with the increase in age [1,28]. The decrease in E wave velocity normally results in increased A wave velocity, thus also resulting in decreased E/A, as observed in this study [1].

TDI-derived E/E′ was demonstrated to be a good indicator of increased LA pressure, caused by volume overload and mitral regurgitation, and an important predictor of congestive heart failure in dogs with myxomatous mitral valve disease [1,29]. The MPI, also obtained via TDI, is a comprehensive index of global myocardial function, which was demonstrated to correlate well with both systolic and diastolic cardiac performance in dogs [1,30]. There are reports indicating that TDI-derived myocardial velocities and LV MPI are influenced by age [1,30,31]. On the contrary, E/E′ and the MPI in this study showed no significant associations with breed, age, nor BW.

The 2D-STE analysis is based on B-mode imaging; it is thus considered angle independent, which allows evaluation to be performed along multiple spatial orientations, and also independent of cardiac translational movement and tethering [3]. It is gaining growing importance in both human and veterinary medicine for the evaluation of myocardial function. The majority of the reported normal values of strain parameters in dogs are of Beagles and mix breeds [32,33,34,35,36,37]. The strain values obtained in this study did not show any significant breed differences, and they were comparable to those in previous reports of other breeds. In terms of Retrievers, Carnabuci et al. reported strain values in healthy LR [38]. While this study used the right parasternal short-axis view at the level of the papillary muscle and GCS was obtained as a transmural strain value, Carnabuci et al. used the right parasternal short-axis view at the basal and apical levels, and endocardial and epicardial GCS were obtained separately. The mid-point value of basal and apical GRS and the mid-point value of basal endocardial and epicardial GCS and apical endocardial and epicardial GCS are comparable to the GRS and GCS values of this study. It was reported that GRS had greater variability than global longitudinal strain and GCS, which was thought to be technical rather than biological [39,40]. Similarly, GRS in this study had much greater standard deviation than GCS; therefore, strain values obtained using the circumferential direction may be more favorable.

LV dyssynchrony, which is the uncoordinated contraction of LV, results in compromised global LV systolic function and energy insufficiency, and is widely recognized as a major contributor of morbidity and mortality in human with heart failure [41]. The STI value in this study was much higher with greater variability than those in previous reports, which could be due to the small sample size.

There are conflicting reports on the association between the strain values and age [36,40,42]. The results of this study showed a significant correlation between GRS and age, but not between GCS and the STI. No significant correlations were observed between the strain parameters and BW, which was coherent with a previous report [32].

There were several limitations that should be mentioned. Firstly, the small sample size of the Retrievers could have affected the results of this study, in terms of, but not limited to, the association between the echocardiographic parameters and variables including breed, age, and BW. Especially, the results of 2D-STE may be less convincing since less than ten dogs were included in each breed of Retrievers. Secondly, the strain analysis was not performed in the longitudinal direction. Strain analyses in the radial and circumferential directions were used in this study, since GRS and GCS analyses can easily be performed even in enlarged LVs, for example, in the case of DCM, whereas the analysis can be challenging in the longitudinal direction, as enlarged LV may move out of the view and may not be traced accurately. However, there are reports that longitudinal strain may be a more sensitive indicator of myocardial dysfunction and may be suitable for the early detection of myocardial dysfunction [43,44]. Further studies in a larger population are warranted to confirm the findings of this study.

The results of this study suggested that while LV dimensions were dependent on BW, the differences between breeds with similar body conformation were minimal and likely to be clinically irrelevant. It is important to note that functional parameters such as EF and FS were independent of age. Therefore, a decrease in these parameters in older Retrievers should be evaluated carefully, since it may indicate the early stage of cardiac diseases such as DCM, which are more commonly observed in aged dogs. Age-dependent parameters included PA Vmax, E, and E/A, indicating age-related reductions in cardiac function, such as reduced vascular elasticity and relaxation. In Retrievers, the parameters of 2D-STE were independent of breed, age, and BW, with the exception of GRS.

## Figures and Tables

**Figure 1 vetsci-09-00529-f001:**
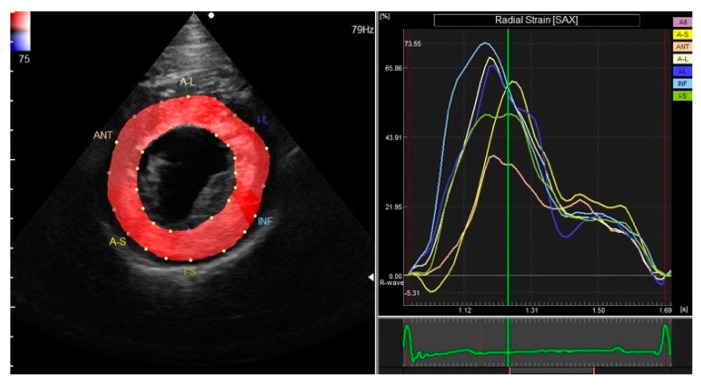
Global peak strain in the radial direction obtained via right parasternal short-axis view at the level of the papillary muscle.

**Figure 2 vetsci-09-00529-f002:**
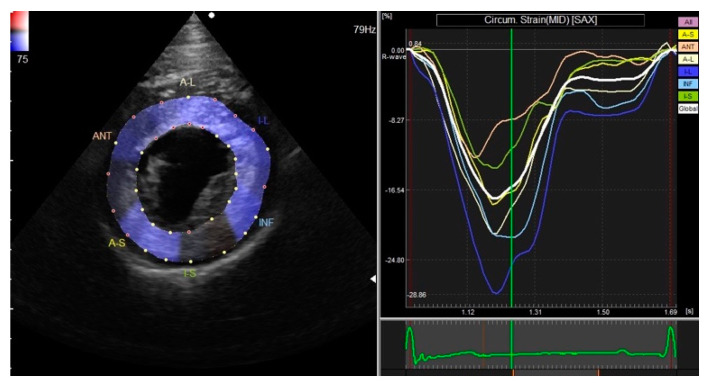
Global peak strain in the circumferential direction obtained via right parasternal short-axis view at the level of the papillary muscle.

**Table 1 vetsci-09-00529-t001:** Standard echocardiographic parameters of three clinically healthy Retriever breeds.

	Flat-Coated (*n* = 16)	Golden (*n* = 16)	Labrador (*n* = 14)	*p*-Value
**IVSd (mm), mean ± SD**	8.59 ± 0.93	10.06 ± 1.17	9.60 ± 1.95	**0.0062**
**IVSs (mm), mean ± SD**	11.82 ± 1.37	14.21 ± 1.45	13.91 ± 2.52	**0.0007**
**LVIDd (mm), mean ± SD**	40.70 ± 2.93	39.89 ± 3.43	40.61 ± 2.91	0.8038
**LVIDs (mm), mean ± SD**	28.42 ± 2.34	26.67 ± 3.15	27.93 ± 2.49	0.1686
**LVFWd (mm), median (range)**	8.00 (6.80–9.60)	10.70 (8.00–15.33)	8.88 (7.00–14.00)	**<0.0001**
**LVFWs (mm), median (range)**	11.30 (9.80–13.00)	15.00 (11.80–19.40)	12.55 (10.00–21.50)	**<0.0001**
**LVIDdN, median (range)**	1.60 (1.30–1.67)	1.49 (1.30–1.69)	1.53 (1.41–1.69)	0.1888
**LVIDsN, mean ± SD**	1.02 ± 0.08	0.93 ± 0.10	0.99 ± 0.06	**0.0249**
**EF (%), mean ± SD**	65.42 ± 4.75	69.30 ± 6.73	67.52 ± 4.72	0.2064
**FS (%), median (range)**	29.62 (26.24–37.80)	32.77 (25.63–41.00)	32.23 (26.76–36.36)	0.2085
**LA/Ao, mean ± SD**	1.29 ± 0.10	1.33 ± 0.09	1.23 ± 0.10	0.1040
**PA Vmax (cm/s), mean ± SD**	81.67 ± 13.11	80.00 ± 11.91	78.32 ± 17.51	0.8366
**Ao Vmax (cm/s), mean ± SD**	84.54 ± 17.71	97.86 ± 14.60	95.92 ± 22.23	0.1574
**E (cm/s), median (range)**	63.43 (32.46–94.10)	52.83 (43.54–85.30)	53.11 (45.48–91.46)	0.5767
**A (cm/s), median (range)**	49.03 (39.34–68.20)	49.05 (24.36–64.60)	43.04 (35.68–73.30)	0.1803
**E/A, mean ± SD**	1.25 ± 0.37	1.24 ± 0.33	1.35 ± 0.32	0.7465
**E/E′ IVS, mean ± SD**	6.60 ± 1.60	7.40 ± 1.80	6.80 ± 2.30	0.5402
**E/E′ LVFW, median (range)**	3.84 (2.78–10.07)	5.04 (3.40–8.90)	4.52 (2.58–6.60)	0.2421
**MPI IVS, mean ± SD**	0.87 ± 0.24	0.66 ± 0.21	0.97 ± 0.37	0.0508
**MPI LVFW, mean ± SD**	0.87 ± 0.26	0.75 ± 0.27	1.01 ± 0.40	0.1882

SD, standard deviation; IVSd, interventricular septal thickness at end-diastole; IVSs, interventricular septal thickness at end-systole; LVIDd, left ventricular internal diameter at end-diastole; LVIDs, left ventricular internal diameter at end-systole; LVFWd, left ventricular free wall thickness at end-diastole; LVFWs, left ventricular free wall thickness at end-systole; LVIDdN, diastolic left ventricular internal diameter normalized to body weight; LVIDsN, systolic left ventricular internal diameter normalized to body weight; EF, ejection fraction; FS, fractional shortening; LA/Ao, left-atrial-to-aortic ratio; PA Vmax, peak pulmonary blood flow velocity; Ao Vmax, peak aortic blood flow velocity; E, rapid ventricular filling; A, atrial contraction; E/A, ratio of rapid ventricular filling to atrial contraction; E/E′ IVS, ratio of rapid ventricular filling to rapid diastolic motion at interventricular septum; E/E′ LVPW, ratio of rapid ventricular filling to rapid diastolic motion at left ventricular free wall; MPI IVS, myocardial performance index at interventricular septum; MPI LVPW, myocardial performance index at left ventricular free wall.

**Table 2 vetsci-09-00529-t002:** Standard echocardiographic parameters of clinically healthy Retrievers.

	Retrievers (*n* = 46)			
	Mean ± SD	Median	95% CI of Mean	Range
**IVSd (mm)**	9.41 ± 1.50	9.30	8.96–9.85	6.80–13.95
**IVSs (mm)**	13.29 ± 2.09	13.20	12.67–13.91	9.75–19.00
**LVIDd (mm)**	40.39 ± 3.06	40.80	39.48–41.30	33.00–45.00
**LVIDs (mm)**	27.66 ± 2.74	27.95	26.85–28.48	21.60–32.00
**LVFWd (mm)**	9.52 ± 2.03	9.10	8.92–10.13	6.80–15.33
**LVFWs (mm)**	13.24 ± 2.75	12.38	12.42–14.05	9.80–21.50
**LVIDdN**	1.53 ± 0.11	1.54	1.50–1.56	1.30–1.69
**LVIDsN**	0.98 ± 0.09	1.00	0.95–1.00	0.80–1.12
**EF (%)**	67.41 ± 5.65	67.66	65.71–69.10	58.87–78.84
**FS (%)**	31.48 ± 4.17	30.89	30.25–32.72	25.63–41.00
**LA/Ao**	1.28 ± 0.10	1.27	1.24–1.32	1.08–1.47
**PA Vmax (cm/s)**	80.11 ± 13.96	79.66	75.82–84.41	52.30–103.20
**Ao Vmax (cm/s)**	92.14 ± 19.12	90.95	86.18–98.10	60.45–125.90
**E (cm/s)**	58.68 ± 14.04	54.27	54.52–62.85	32.46–94.10
**A (cm/s)**	47.82 ± 9.52	48.04	44.99–50.65	24.36–73.30
**E/A**	1.28 ± 0.33	1.26	1.18–1.38	0.75–2.05
**E/E′ IVS**	6.97 ± 1.89	6.90	6.40–7.55	3.28–11.55
**E/E′ LVFW**	4.99 ± 1.69	4.57	4.46–5.52	2.58–10.07
**MPI IVS**	0.83 ± 0.30	0.81	0.73–0.94	0.30–1.62
**MPI LVFW**	0.88 ± 0.32	0.79	0.76–0.99	0.38–1.64

SD, standard deviation; CI, confidence interval; IVSd, interventricular septal thickness at end-diastole; IVSs, interventricular septal thickness at end-systole; LVIDd, left ventricular internal diameter at end-diastole; LVIDs, left ventricular internal diameter at end-systole; LVFWd, left ventricular free wall thickness at end-diastole; LVFWs, left ventricular free wall thickness at end-systole; LVIDdN, diastolic left ventricular internal diameter normalized to body weight; LVIDsN, systolic left ventricular internal diameter normalized to body weight; EF, ejection fraction; FS, fractional shortening; LA/Ao, left-atrial-to-aortic ratio; PA Vmax, peak pulmonary blood flow velocity; Ao Vmax, peak aortic blood flow velocity; E, rapid ventricular filling; A, atrial contraction; E/A, ratio of rapid ventricular filling to atrial contraction; E/E′ IVS, ratio of rapid ventricular filling to rapid diastolic motion at interventricular septum; E/E′ LVPW, ratio of rapid ventricular filling to rapid diastolic motion at left ventricular free wall; MPI IVS, myocardial performance index at interventricular septum; MPI LVPW, myocardial performance index at left ventricular free wall.

**Table 3 vetsci-09-00529-t003:** Two-dimensional speckle-tracking echocardiographic parameters of three clinically healthy Retriever breeds.

	Flat-Coated (*n* = 10)	Golden (*n* = 8)	Labrador (*n* = 10)	*p*-Value
**GRS**	41.37 ± 7.33	39.81 ± 14.02	37.29 ± 15.80	0.6610
**GCS**	−12.24 ± 1.61	−13.81± 3.54	−13.09 ± 2.14	0.2826
**STI**	57.56 ± 21.35	72.92 ± 18.35	68.19 ± 31.58	0.4528

SD, standard deviation; GRS, global peak radial strain; STI, synchrony time index; GCS, global peak circumferential strain.

**Table 4 vetsci-09-00529-t004:** Two-dimensional speckle-tracking echocardiographic parameters of clinically healthy Retrievers.

	Retrievers (*n* = 28)			
	Mean ± SD	Median	95% CI of Mean	Range
**GRS**	39.46 ± 12.46	37.93	34.63–44.29	19.46–66.46
**GCS**	−12.99 ± 2.46	−12.92	−13.94–−12.04	−19.40–−9.32
**STI**	65.74 ± 24.78	62.20	56.13–75.35	24.58–119.00

SD, standard deviation; CI, confidence interval; GRS, global peak radial strain; STI, synchrony time index; GCS, global peak circumferential strain.

## Data Availability

The data presented in this study are available on request.

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
