# Peer review of "Evaluation of Left Ventricular Function in Healthy Retrievers Using Standard and 2D Speckle-Tracking Echocardiography"

_vetsci, 2022, doi:10.3390/vetsci9100529_

Round 1

Reviewer 1 Report

The study compares standard and 2D-STE-derived echocardiographic parameters between three different breeds of retrievers and evaluate the effect of some independent variables on echocardiographic parameters. The study is well planned although the small sample size may have a significant effect on the results and discussion should be expanded and enriched with comparisons with previous studies. Despite these limitations, I believe that the manuscript can be published if the following minor revisions are made:

Line 29: please replace “contraction influenced by age” with “were influenced by age”.

Lines 29-30: Please replace “In 2D-STE, the strain parameters were independent of breed, age, and BW, except for global stain in the radial direction” with “The 2D-STE-derived radial and circumferential strain parameters were independent of breed, age, and BW, except for global stain in the radial direction”. 

Linea 60: Please replace “Aim of the present study is” with “Aim of the present study was”

Lines 98-115 please provide references.

Line 131: please delete the word “Significant” and start the sentence with “Differences of…”

Lines 141-148: it would be more correct to indicate the normally distributed variables as mean ± standard deviation (and eventually range) and those not normally distributed as median and range.

Tables 1, 2 and 3: I suggest deleting tables 1,2 and 3 and report their results in a single table in which normally distributed variables are indicated with mean and SD and non-normally distributed variables with median and range. In this way it is easier for the reader to understand if there are significant differences and which of the breeds studied has higher or lower values of each variable without having to look at 3 different tables.

Below is an example:

FC (n=16)

GR (n=16)

LR (n=14)

P value

Variable (mm), mean (SD)

1(1)

2(2)

3(3)

4

Variable (mm), median (range)

1 (0-2)

2 (1-3)

3(2-4)

5

Variable (%), mean (SD)

Variable (%), median (range)

TABLE 4: from this table it is not clear whether 95% CI refer to the mean or the median, please specify.

Lines 166-169: The results of the multiple linear regression should indicate, in addition to the p-value, also the regression coefficient Beta (positive or negative), and the 95% CI of Beta. Please add the missing data. 

Tables 5, 6 and 7: I suggest deleting tables 5,6 and 7 and report their results in a single table in which normally distributed variables are indicated with mean and SD and non-normally distributed variables with median and range. In this way it is easier for the reader to understand if there are significant differences or not and which of the breeds studied has higher or lower values of each variable without having to look at 3 different tables.

Below is an example:

FC (n=10)

GR (n=8)

LR (n=10)

P value

Variable (mm), mean (SD)

1(1)

2(2)

3(3)

4

Variable (mm), median (range)

1 (0-2)

2 (1-3)

3(2-4)

5

Variable (%), mean (SD)

Variable (%), median (range)

TABLE 8: from this table it is not clear whether 95% CI refer to the mean or the median, please specify.

Line 190: From tables 5,6,7 and 8 it is evident that the dogs who underwent to 2D-STE examination were 28 out of 46. This should be specified in the text of results section.

Lines 193-194: The results of the multiple linear regression should indicate, in addition to the p-value, also the regression coefficient Beta (positive or negative), and the 95% CI of Beta. Please add the missing data.

Lines 257-254: Are your results similar to those found by other authors on the same breeds? Please discuss it. (Eg: Carnabucci et al 2013: Assessment of cardiac function using global and regional left ventricular endomyocardial and epimyocardial peak systolic strain and strain rate in healthy Labrador retriever dogs)

Lines 275-277: have you formulated any hypotheses that can explain this result? Was this result also found by other authors in other studies?

Lines 257-274: You should argue more why you have decided to evaluate only the radial and circumferential strain leaving out the longitudinal one. In which studies has the clinical utility of radial and circumferential strain been demonstrated? 2D-STE-derived longitudinal strain may be more sensitive in detecting changes in myocardial contractility in patients with systemic diseases (Eg: Corda et al 2019: Use of 2-dimensional speckle-tracking echocardiography to assess left ventricular systolic function in dogs with systemic inflammatory response syndrome; Hamabe et al 2022: Role of Two-Dimensional Speckle-Tracking Echocardiography in Early Detection of Left Ventricular Dysfunction in Dogs)

Please discuss these topics.

Lines 279-284: the limitations section should be expanded: the small sample size could in fact have influenced all the results of this study, not only those relating to the relationships between variables. In particular, the results of 2D-STE are weak because they were performed on 28 subjects divided into 3 groups. 

Reviewer 2 Report

The article is written in a clear, transparent way, interesting for the readers.

In line 117 correct capital letter- “two” “Two-dimensional”.

Reviewer 3 Report

-49: I would use 'is it' instead of 'it is'.

-59: I would put the word 'the' in front of the word 'breed'

-61: 'A type'

-62: I would drop 'without damage'

-64-65: I would rephrase this sentence. 

-91: What is meant by 'optimal air condition'? Temperature? I would specify or drop. 

-General comment 1: the FR group was clearly younger than the GR group, and this should be taken into account when comparing both breeds. I would advise to mention this or comment/elaborate on this in the discussion.

-General comment 2: If possible, I would add a figure of an echocardiographic image for illustrative purposes.

Overall good work, congratulations!
